# Heterogeneous Bitwidth Binarization in Convolutional Neural Networks

**Josh Fromm**
Department of Electrical Engineering
University of Washington
Seattle, WA 98195
jwfromm@uw.edu

**Shwetak Patel**
Department of Computer Science
University of Washington
Seattle, WA 98195
shwetak@cs.washington.edu

**Matthai Philipose**
Microsoft Research
Redmond, WA 98052
matthaip@microsoft.com

## Abstract

Recent work has shown that fast, compact low-bitwidth neural networks can be surprisingly accurate. These networks use *homogeneous binarization*: all parameters in each layer or (more commonly) the whole model have the same low bitwidth (e.g., 2 bits). However, modern hardware allows efficient designs where each arithmetic instruction can have a custom bitwidth, motivating *heterogeneous binarization*, where every parameter in the network may have a different bitwidth. In this paper, we show that it is feasible and useful to select bitwidths at the parameter granularity during training. For instance a heterogeneously quantized version of modern networks such as AlexNet and MobileNet, with the right mix of 1-, 2- and 3-bit parameters that average to just 1.4 bits can equal the accuracy of homogeneous 2-bit versions of these networks. Further, we provide analyses to show that the heterogeneously binarized systems yield FPGA- and ASIC-based implementations that are correspondingly more efficient in both circuit area and energy efficiency than their homogeneous counterparts.

## 1 Introduction

With Convolutional Neural Networks (CNNs) now outperforming humans in vision classification tasks (Szegedy et al., 2015), it is clear that CNNs will be a mainstay of AI applications. However, CNNs are known to be computationally demanding, and are most comfortably run on GPUs. For execution in mobile and embedded settings, or when a given CNN is evaluated many times, using a GPU may be too costly. The search for inexpensive variants of CNNs has yielded techniques such as hashing (Chen et al., 2015), vector quantization (Gong et al., 2014), and pruning (Han et al., 2015). One particularly promising track is binarization (Courbariaux et al., 2015), which replaces 32-bit floating point values with single bits, either +1 or –1, and (optionally) replaces floating point multiplies with packed bitwise popcount-xnors Hubara et al. (2016). Binarization can reduce the size of models by up to $32\times$, and reduce the number of operations executed by up to $64\times$.

It has not escaped hardware designers that the popcount-xnor operations used in a binary network are especially well suited for FPGAs or ASICs. Taking the xnor of two bits requires a single logic gate compared to the hundreds required for even the most efficient floating point multiplication units (Ehliar, 2014). The drastically reduced area requirements allows binary networks to be implemented with fully parallel computations on even relatively inexpensive FPGAs (Umuroglu et al.,

2017). The level of parallelization afforded by these custom implementations allows them to outperform GPU computation while expending a fraction of the power, which offers a promising avenue of moving state of the art architectures to embedded environments. We seek to improve the occupancy, power, and/or accuracy of these solutions.

Our approach is based on the simple observation that the power consumption, space needed, and accuracy of binary models on FPGAs and custom hardware are proportional to $mn$, where $m$ is the number of bits used to binarize input activations and $n$ is the number of bits used to binarize weights. Current binary algorithms restrict $m$ and $n$ to be integer values, in large part because efficient CPU implementations require parameters within a layer to be the same bitwidth. However, hardware has no such requirements. Thus, we ask whether bitwidths can be fractional. To address this question, we introduce Heterogeneous Bitwidth Neural Networks (HBNNs), which allow *each individual parameter to have its own bitwidth*, giving a fractional average bitwidth to the model.

Our main contributions are:

(1) We propose the problem of selecting the bitwidth of individual parameters during training such that the bitwidths average out to a specified value.

(2) We show how to augment a state-of-the-art homogeneous binarization training scheme with a greedy bitwidth selection technique (which we call "middle-out") and a simple hyperparameter search to produce good heterogeneous binarizations efficiently.

(3) We present a rigorous empirical evaluation (including on highly optimized modern networks such as Google's MobileNet) to show that heterogeneity yields equivalent accuracy at significantly lower average bitwidth.

(4) Although implementing HBNNs efficiently on CPU/GPU may be difficult, we provide estimates based on recently proposed FPGA/ASIC implementations that HBNNs' lower average bitwidths can translate to significant reductions in circuit area and power.

## 2 Homogeneous Network Binarization

In this section we discuss existing techniques for binarization. Table 1 summarizes their accuracy.[1]

When training a binary network, all techniques including ours maintain weights in floating point format. During forward propagation, the weights (and activations, if both weights and activations are to be binarized) are passed through a *binarization function $\mathcal{B}$*, which projects incoming values to a small, discrete set. In backward propagation, a *custom gradient*, which updates the floating point weights, is applied to the binarization layer. After training is complete, the binarization function is applied one last time to the floating point weights to create a true binary (or more generally, small, discrete) set of weights, which is used for inference from then on.

Binarization was first introduced by Courbariaux et al. (2015). In this initial investigation, dubbed BinaryConnect, 32-bit tensors $T$ were converted to 1-bit variants $T^B$ using the stochastic equation

$$\mathcal{B}(T) \triangleq T^B = \begin{cases} +1 & \text{with probability } p = \sigma(T), \\ -1 & \text{with probability } 1 - p \end{cases} \tag{1}$$

where $\sigma$ is the hard sigmoid function defined by $\sigma(x) = \max(0, \min(1, \frac{x+1}{2}))$. For the custom gradient function, BinaryConnect simply used $\frac{dT^B}{dT} = 1$.

Although BinaryConnect showed excellent results on relatively simple datasets such as CIFAR-10 and MNIST, it performed poorly on ImageNet, achieving only an accuracy of 27.9%. Courbariaux et al. (2016) later improved this model by simplifying the binarization by simply taking $T^B = \text{sign}(T)$ and adding a gradient for this operation, namely the *straight-through estimator*:

$$\frac{dT^B}{dT} = 1_{|T| \leq 1}. \tag{2}$$

The authors showed that the straight-through estimator allowed the binarization of activations as well as weights without collapse of model performance. However, they did not attempt to train a model on ImageNet in this work.

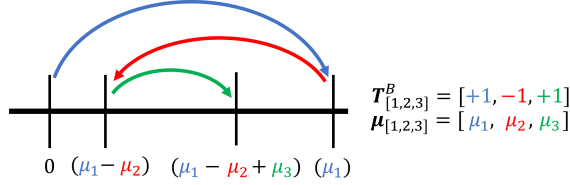

Figure 1: Residual error binarization with $n = 3$ bits. Computing each bit takes a step from the position of the previous bit (see Equation 4).

Rastegari et al. (2016) made a slight modification to the simple pure single bit representation that showed improved results. Now taking a binarized approximation as

$$T^B = \alpha_i \text{sign}(T) \quad \text{with} \quad \alpha_i = \frac{1}{d} \sum_{j=1}^{d} |T_j|. \tag{3}$$

This additional scalar term allows binarized values to better fit the distribution of the incoming floating-point values, giving a higher fidelity approximation for very little extra computation. The addition of scalars and the straight-through estimator gradient allowed the authors to achieve a Top-1 accuracy of 44.2% on ImageNet.

Hubara et al. (2016) and Zhou et al. (2016) found that increasing the number of bits used to quantize the activations of the network gave a considerable boost to the accuracy, achieving similar Top-1 accuracy of 51.03% and 50.7% respectively. The precise binarization function varied, but the typical approaches include linearly or logarithmically placing the quantization points between 0 and 1, clamping values below a threshold distance from zero to zero (Li et al., 2016), and computing higher bits by measuring the residual error from lower bits (Tang et al., 2017). All $n$-bit binarization schemes require similar amounts of computation at inference time, and have similar accuracy (see Table 1). In this work, we extend the *residual error binarization* function Tang et al. (2017) for binarizing to multiple ($n$) bits:

$$T_1^B = \text{sign}(T), \ \mu_1 = \text{mean}(|T|)$$

$$E_n = T - \sum_{i=1}^{n} \mu_i \times T_i^B$$

$$T_{n>1}^B = \text{sign}(E_{n-1}), \ \mu_{n>1} = \text{mean}(|E_{n-1}|) \tag{4}$$

$$T \approx \sum_{i=1}^{n} \mu_i \times T_i^B$$

where $T$ is the input tensor, $E_n$ is the residual error up to bit $n$, $T_n^B$ is a tensor representing the $n^{\text{th}}$ bit of the approximation, and $\mu_n$ is a scaling factor for the $n^{\text{th}}$ bit. Note that the calculation of bit $n$ is a recursive operation that relies on the values of all bits less than $n$. Residual error binarization has each additional bit take a step from the value of the previous bit. Figure 1 illustrates the process of binarizing a single value to 3 bits. Since every binarized value is derived by taking $n$ steps, where each step goes left or right, residual error binarization approximates inputs using one of $2^n$ values.

## 3 Heterogeneous Binarization

To date, there remains a considerable gap between the performance of 1-bit and 2-bit networks (compare rows 8 and 10 of Table 1). The highest full (i.e., where both weights and activations are quantized) single-bit performer on AlexNet, Xnor-Net, remains roughly 7 percentage points less accurate (top 1) than the 2-bit variant, which is itself about 5.5 points less accurate than the 32-bit variant (row 25). When only weights are binarized, very recent results (Dong et al., 2017) similarly find that binarizing to 2 bits can yield nearly full accuracy (row 2), while the 1-bit equivalent lags by 4 points (row 1). The flip side to using 2 bits for binarization is that the resulting models require double the number of operations as the 1-bit variants at inference time.

These observations naturally lead to the question, explored in this section, of whether it is possible to attain accuracies closer to those of 2-bit models while running at speeds closer to those of 1-bit variants. Of course, it is also fundamentally interesting to understand whether it is possible to match

the accuracy of higher bitwidth models with those that have lower (on average) bitwidth. Below, we discuss how to extend residual error binarization to allow heterogeneous (effectively fractional) bitwidths and present a method for distributing the bits of a heterogeneous approximation.

## 3.1 Heterogeneous Residual Error Binarization via a Mask Tensor

We modify Equation 4 , which binarizes to $n$ bits, to instead binarize to a mixture of bitwidths by changing the third line as follows:

$$T_{n>1}^B = \text{sign}(E_{n-1,j}), \ \mu_{n>1} = \text{mean}(|E_{n-1,j}|) \tag{5}$$
$$\text{with } j : M_j \geq n$$

Note that the only addition is the *mask tensor* $M$, which is the same shape as $T$, and specifies the number of bits $M_j$ that the $j$<sup>th</sup> entry of $T$ should be binarized to. In each round $n$ of the binarization recurrence, we now only consider values that are not finished binarizing, i.e, which have $M_j \geq n$. Unlike homogeneous binarization, therefore, heterogeneous binarization generates binarized values by taking *up to*, not necessarily exactly, $n$ steps. Thus, the number of distinct values representable is $\sum_{i=1}^{n} 2^i = 2^{n+1} - 2$, which is roughly double that of the homogeneous binarization.

In the homogeneous case, on average, each step improves the accuracy of the approximation, but there may be certain individual values that would benefit from not taking a step, in Figure 1 for example, it is possible that $(\mu_1 - \mu_2)$ approximates the target value better than $(\mu_1 - \mu_2 + \mu_3)$. If values that benefit from not taking a step can be targeted and assigned fewer bits, the overall approximation accuracy will improve despite there being a lower average bitwidth.

## 3.2 Computing the Mask Tensor $M$

The question of how to distribute bits in a heterogeneous binary tensor to achieve high representational power is equivalent to asking how $M$ should be generated. When computing $M$, our goal is to take an average bitwidth $B$ and determine both what fraction $P$ of $M$ should be binarized to each bitwidth (e.g., $P = 5\%$ 3-bit, 10% 2-bit and 85% 1-bit for an average of $B = 1.2$ bits), and how to distribute these bitwidths across the individual entries in $M$. The full computation of $M$ is described in Algorithm 1.

We treat the distribution $P$ over bitwidths as a model-wide hyperparameter. Since we only search up to 3 bitwidths in practice, we perform a simple grid sweep over the values of $P$. As we discuss in Section 4.3, our discretization is relatively insensitive to these hyperparameters, so a coarse sweep is adequate. The results of the sweep are represented by the function $DistFromAvg$ in Algorithm 1.

Given $P$, we need to determine how to distribute the various bitwidths using a value aware method: assigning low bitwidths to values that do not need additional approximation and high bitwidths to

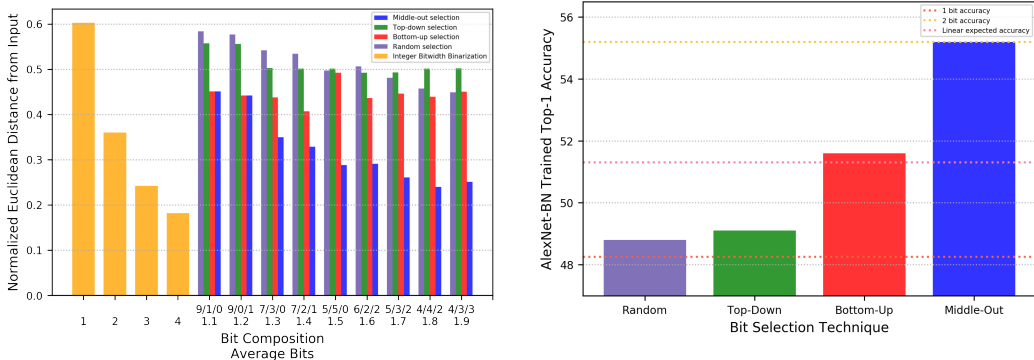

(a) Bit selection representational power.

(b) 1.4 bit HBNN AlexNet Accuracy.

Figure 2: Effectiveness of heterogeneous bit selection techniques (a) ability of different binarization schemes to approximate a large tensor of normally distributed random values. (b) accuracy of 1.4 bit heterogeneous binarized AlexNet-BN trained using each bit-selection technique.

**Algorithm 1** Generation of bit map $M$.

---

**Input:** A tensor $T$ of size $N$ and an average bitwidth $B$.
**Output:** A bit map $M$ that can be used in Equation 5 to heterogeneously binarize $T$.

1:   $R = T$           $\triangleright$ Initialize $R$, which contains values that have not yet been assigned a bitwidth
2:   $x = 0$
3:   $P = \text{DistFromAvg}(B)$                   $\triangleright$ Generate distribution of bits to fit average.
4:   **for** $(b, p_b)$ in $P$ **do**       $\triangleright$ $b$ is a bitwidth and $p_b$ is the percentage of $T$ to binarize to width $b$.
5:      $S = \text{SortHeuristic}(R)$ $\triangleright$ Sort indices of remaining values by suitability for $b$-bit binarization.
6:      $M[S[x : x + p_b N]] = b$
7:      $R = R \setminus R[S[x : x + p_b N]]$           $\triangleright$ Do not consider these indices in next step.
8:      $x \mathrel{+}= p_b N$
9:   **end for**

---

those that do. To this end, we propose several sorting heuristic methods: Top-Down (TD), Middle-Out (MO), Bottom-Up (BU), and Random (R). These methods all attempt to sort values of $T$ based on how many bits that value should be binarized with. For example, Top-Down sorting assumes that larger values need fewer bits, and so performs a standard descending sort. Similarly, Middle-Out sorting distributes fewer bits to values closest to the mean of $T$, while Bottom-Up sorting assigns fewer bits to smaller values. As a simple we control, we also consider Random sorting, which assigns bits in a completely uninformed way. The definitions for the sorting heuristics is given by Equation 6.

$$
\begin{aligned}
\text{TD}(T) &= \text{sort}(|T|, \text{descending}) \\
\text{MO}(T) &= \text{sort}(|T| - \text{mean}(|T|), \text{ascending}) \\
\text{BU}(T) &= \text{sort}(|T|, \text{ascending}) \\
\text{R}(T) &= \text{a fixed uniformly random permutation of } T
\end{aligned}
\tag{6}
$$

To evaluate the methods in Equation 6, we performed two experiments. In the first, we create a large tensor of normally distributed values and binarize it with a variety of bit distributions $P$ and each of the sorting heuristics using Algorithm 1. We then computed the Euclidean distance between the binarized tensor and the original full precision tensor. A lower normalized distance suggests a more powerful sorting heuristic. The results of this experiment are shown in Figure 2a, and show that Middle-Out sorting outperforms other heuristics by a significant margin. Notably, the results suggest that using Middle-Out sorting can produce approximations with fewer than 2-bits that are comparably accurate to 3-bit integer binarization.

To confirm these results translate to accuracy in binarized convolutional networks, we consider 1.4 bit binarized AlexNet, with bit distribution $P$ set to 70% 1-bit, 20% 2-bit, and 10% 3-bit, an average of 1.4 bits. The specifics of the model and training procedure are the same as those described in Section 4.1. We train this model with each of the sorting heuristics and compare the final accuracy to gauge the representational strength of each heuristic. The results are shown in Figure 2b. As expected, Middle-Out sorting performs significantly better than other heuristics and yields an accuracy comparable to 2-bit integer binarization despite using on average 1.4 bits.

The intuition behind the exceptional performance of Middle-Out is based on Figure 1 . We can see that the values that are most likely to be accurate without additional bits are those that are closest to the average $\mu_n$ for each step $n$. By assigning low bitwidths to the most average values, we can not just minimize losses, but in some cases provide a better approximation using fewer average steps. In proceeding sections, all training and evaluation is performed with Middle-Out as the sorting heuristic in Algorithm 1.

## 4   Experiments

To evaluate HBNNs we wished to answer the following three questions:

    (1)   How does accuracy scale with an uninformed bit distribution?

    (2)   How well do HBNNs perform on a challenging dataset compared to the state of the art?

    (3)   Can the benefits of HBNNs be transferred to other architectures?

In this section we address each of these questions.

## 4.1    Implementation Details

AlexNet with batch-normalization (AlexNet-BN) is the standard model used in binarization work due to its longevity and the general acceptance that improvements made to accuracy transfer well to more modern architectures. Batch normalization layers are applied to the output of each convolution block, but the model is otherwise identical to the original AlexNet model proposed by Krizhevsky et al. (2012). Besides it's benefits in improving convergence, Rastegari et al. (2016) found that batch-normalization is especially important for binary networks because of the need to equally distribute values around zero. We additionally insert binarization functions within the convolutional layers of the network when binarizing weights and at the input of convolutional layers when binarizing inputs. We keep a floating point copy of the weights that is updated during back-propagation, and binarized during forward propagation as is standard for binary network training. We use the straight-through estimator for gradients.

When binarizing the weights of the network's output layer, we add a single parameter scaling layer that helps reduce the numerically large outputs of a binary layer to a size more amenable to softmax, as suggested by Tang et al. (2017). We train all models using an SGD solver with learning rate 0.01, momentum 0.9, and weight decay 1e-4 and randomly initialized weights for 90 epochs on PyTorch.

## 4.2    Layer-level Heterogeneity

As a baseline, we test a "poor man's" approach to HBNNs, where we fix up front the number of bits each layer is allowed, require all values in a layer to have its associated bitwidth, and then train as with conventional homogeneous binarization. We consider 10 mixes of 1, 2 and 3-bit layers so as to sweep average bitwidths between 1 and 2. We trained as described in Section 4.1. For this experiment, we used the CIFAR-10 dataset with a deliberately hobbled (4-layer fully convolutional) model with a maximum accuracy of roughly 78% as the baseline 32-bit variant. We chose CIFAR-10 to allow quick experimentation. We chose not to use a large model for CIFAR-10, because for large models it is known that even 1-bit models have 32-bit-level accuracy Courbariaux et al. (2016).

Figure 3a shows the results. Essentially, accuracy increases roughly linearly with average bitwidth. Although such linear scaling of accuracy with bitwidth is itself potentially useful (since it allows finer grain tuning on FPGAs), we are hoping for even better scaling with the "data-aware" bitwidth selection provided by HBNNs.

## 4.3    Bit Distribution Generation

As described in 3.2, one of the considerations when using HBNNs is how to take a desired average bitwidth and produce a matching distribution of bits. For example, using 70% 1-bit, 20% 2-bit and

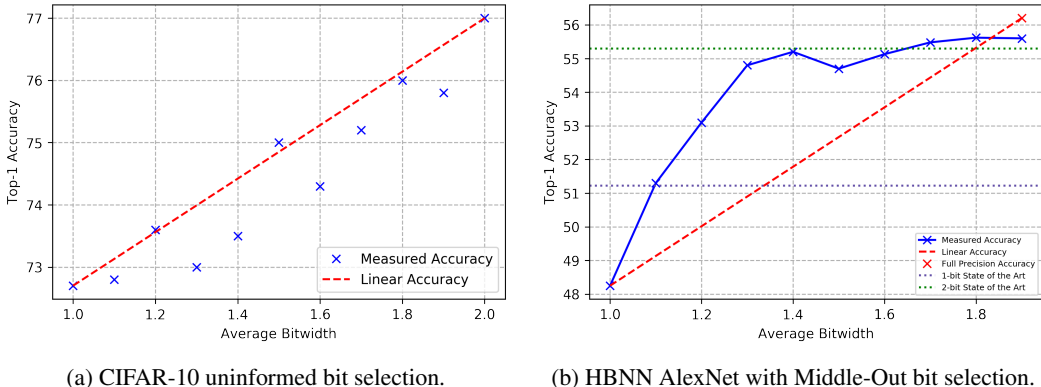

(a) CIFAR-10 uninformed bit selection.          (b) HBNN AlexNet with Middle-Out bit selection.

Figure 3: Accuracy results of trained HBNN models. (a) Sweep of heterogenous bitwidths on a deliberately simplified four layer convolutional model for CIFAR-10. (b) Accuracy of heterogeneous bitwidth AlexNet-BN models. Bits are distributed using the Middle-Out selection algorithm.

Table 1: Accuracy of related binarization work and our results

| | Model | Name | Binarization (Inputs / Weights) | Top-1 | Top-5 |
|---|---|---|---|---|---|
| | | | Binarized weights with floating point activations | | |
| 1 | AlexNet | SQ-BWN (Dong et al., 2017) | full precision / 1-bit | 51.2% | 75.1% |
| 2 | AlexNet | SQ-TWN (Dong et al., 2017) | full precision / 2-bit | 55.3% | 78.6% |
| 3 | AlexNet | TWN (our implementation) | full precision / 1-bit | 48.3% | 71.4% |
| 4 | AlexNet | TWN | full precision / 2-bit | 54.2% | 77.9% |
| 5 | AlexNet | HBNN (our results) | full precision / 1.4-bit | 55.2% | 78.4% |
| 6 | MobileNet | HBNN | full precision / 1.4-bit | 65.1% | 87.2% |
| | | | Binarized weights and activations excluding input and output layers | | |
| 7 | AlexNet | BNN (Courbariaux et al., 2015) | 1-bit / 1-bit | 27.9% | 50.4% |
| 8 | AlexNet | Xnor-Net (Rastegari et al., 2016) | 1-bit / 1-bit | 44.2% | 69.2% |
| 9 | AlexNet | DoReFaNet (Zhou et al., 2016) | 2-bit / 1-bit | 50.7% | 72.6% |
| 10 | AlexNet | QNN (Hubara et al., 2016) | 2-bit / 1-bit | 51.0% | 73.7% |
| 11 | AlexNet | our implementation | 2-bit / 2-bit | 52.2% | 74.5% |
| 12 | AlexNet | our implementation | 3-bit / 3-bit | 54.2% | 78.1% |
| 13 | AlexNet | HBNN | 1.4-bit / 1.4-bit | 53.2% | 77.1% |
| 14 | AlexNet | HBNN | 1-bit / 1.4-bit | 49.4% | 72.1% |
| 15 | AlexNet | HBNN | 1.4-bit / 1-bit | 51.5% | 74.2% |
| 16 | AlexNet | HBNN | 2-bit / 1.4-bit | 52.0% | 74.5% |
| 17 | MobileNet | our implementation | 1-bit / 1-bit | 52.9% | 75.1% |
| 18 | MobileNet | our implementation | 2-bit / 1-bit | 61.3% | 80.1% |
| 19 | MobileNet | our implementation | 2-bit / 2-bit | 63.0% | 81.8% |
| 20 | MobileNet | our implementation | 3-bit / 3-bit | 65.9% | 86.7% |
| 21 | MobileNet | HBNN | 1-bit / 1.4-bit | 60.1% | 78.7% |
| 22 | MobileNet | HBNN | 1.4-bit / 1-bit | 62.0% | 81.3% |
| 23 | MobileNet | HBNN | 1.4-bit / 1.4-bit | 64.7% | 84.9% |
| 24 | MobileNet | HBNN | 2-bit / 1.4-bit | 63.6% | 82.2% |
| | | | Unbinarized (our implementation) | | |
| 25 | AlexNet | (Krizhevsky et al., 2012) | full precision / full precision | 56.5% | 80.1% |
| 26 | MobileNet | (Howard et al., 2017) | full precision / full precision | 68.8% | 89.0% |

10% 3-bit values gives an average of 1.4 bits, but so too does 80% 1-bit and 20% 3-bit values. We suspected that the choice of this distribution would have a significant impact on the accuracy of trained HBNNs, and performed a hyperparameter sweep by varying $DistFromAvg$ in Algorithm 1 when training AlexNet on ImageNet as described in the following sections. However, much to our surprise, **models trained with the same average bitwidth achieved nearly identical accuracies regardless of distribution**. For example, the two 1.4-bit distributions given above yield accuracies of 49.4% and 49.3% respectively. This suggests that choice of $DistFromAvg$ is actually unimportant, which is quite convenient as it simplifies training of HBNNs considerably.

## 4.4 AlexNet: Binarized Weights and Non-Binarized Activations

Recently, Dong et al. (2017) were able to binarize the weights of an AlexNet-BN model to 2 bits and achieve nearly full precision accuracy (row 2 of Table 1). We consider this to be the state of the art in weight binarization since the model achieves excellent accuracy despite all layer weights being binarized, including the input and output layers which have traditionally been difficult to approximate. We perform a sweep of AlexNet-BN models binarized with fractional bitwidths using middle-out selection with the goal of achieving comparable accuracy using fewer than two bits.

The results of this sweep are shown in Figure 3b. We were able to achieve nearly identical top-1 accuracy to the best full 2 bit results (55.3%) with an average of only 1.4 bits (55.2%). As we had hoped, we also found that the accuracy scales in a *super-linear* manner with respect to bitwidth when using middle-out bit selection. Specifically, the model accuracy increases extremely quickly from 1 bit to 1.3 bits before slowly approaching the full precision accuracy.

## 4.5 AlexNet: Binarized Weights and Activations

In order to realize the speed-up benefits of binarization (on CPU or FPGA) in practice, it is necessary to binarize both inputs the weights, which allows floating point multiplies to be replaced with packed bitwise logical operations. The number of operations in a binary network is reduced by a factor of $\frac{64}{mn}$ where $m$ is the number of bits used to binarize inputs and $n$ is the number of bits to binarize weights. Thus, there is significant motivation to keep the bitwidth of both inputs and weights as low as possible without losing too much accuracy. When binarizing inputs, the input and output layers are typically not binarized as the effects on the accuracy are much larger than other layers. We perform another sweep on AlexNet-BN with all layers but the input and output fully binarized and compare the accuracy of HBNNs to several recent results. Row 8 of Table 1 is the top previously reported accuracy (44.2%) for single bit input and weight binarization, while row 10 (51%) is the top accuracy for 2-bit inputs and 1-bit weights.

Table 1 (rows 13 to 16) reports a selection of results from this search. Using 1.4 bits to binarize inputs and weights ($mn = 1.4 \times 1.4 = 1.96$) gives a very high accuracy (53.2% top-1) while having the same number of total operations $mn$ as a network, such as the one from row 10, binarized with 2 bit activations and 1 bit weights. We have similarly good results when leaving the input binarization bitwidth an integer. Using 1 bit inputs and 1.4 bit weights, we reach 49.4% top-1 accuracy which is a large improvement over Rastegari et al. (2016) at a small cost. We found that using more than 1.4 average bits had very little impact on the overall accuracy. Binarizing inputs to 1.4 bits and weights to 1 bit (row 15) similarly outperforms Hubara et al. (2016) (row 10).

## 4.6 MobileNet Evaluation

Although AlexNet serves as an essential measure to compare to previous and related work, it is important to confirm that the benefits of heterogeneous binarization is model independent. To this end, we perform a similar sweep of binarization parameters on MobileNet, a state of the art architecture that has unusually high accuracy for its low number of parameters (Howard et al., 2017). MobileNet is made up of separable convolutions instead of the typical dense convolutions of AlexNet. Each separable convolution is composed of an initial spatial convolution followed by a depth-wise convolution. Because the vast bulk of computation time is spent in the depth-wise convolution, we binarize only its weights, leaving the spatial weights floating point. We binarize the depth wise weights of each MobileNet layer in a similar fashion as in section 4.4 and achieve a Top-1 accuracy of 65.1% (row 6). This is only a few percent below our unbinarized implementation (row 26), which is an excellent result for the significant reduction in model size.

We additionally perform a sweep of many different binarization bitwidths for both the depth-wise weights and input activations of MobileNet, with results shown in rows 17-24 of Table 1. Just as in the AlexNet case, we find that MobileNet with an average of 1.4 bits (rows 21 and 22) achieves over 10% higher accuracy than 1-bit binarization (row 17). We similarly observe that 1.4-bit binarization outperforms 2-bit binarization in each permutation of bitwidths. The excellent performance of HBNN MobileNet confirms that heterogeneous binarization is fundamentally valuable, and we can safely infer that it is applicable to many other network architectures as well.

## 5 Hardware Implementability

Our experiments demonstrate that HBNNs have significant advantages compared to integer bitwidth approximations. However, with these representational benefits come added complexity in implementation. Binarization typically provides a significant speed up by packing bits into 64-bit integers, allowing a CPU or GPU to perform a single xnor operation in lieu of 64 floating-point multiplications. However, Heterogeneous tensors are essentially composed of sparse arrays of bits. Array sparsity makes packing bits inefficient, nullifying much of the speed benefits one would expect from having fewer average bits. The necessity of bit packing exists because CPUs and GPUs are designed to operate on groups of bits rather than individual bits. However, programmable or custom hardware such as FPGAs and ASICs have no such restriction. In hardware, each parameter can have its own set of $n$ xnor-popcount units, where $n$ is the bitwidth of that particular parameter. In FPGAs and ASICs, the total number of computational units in a network has a significant impact on the power consumption and speed of inference. Thus, the benefits of HBNNs, higher accuracy with fewer computational units, are fully realizable.

Table 2: Hardware Implementation Metrics

| | Platform | Model | Unfolding | Bits | Occupancy | kFPS | $P_{chip}$ (W) | Top-1 |
|---|---|---|---|---|---|---|---|---|
| | | | | CIFAR-10 Baseline Implementations | | | | |
| 1 | ZC706 | VGG-8 | 1× | 1 | 21.2% | 21.9 | 3.6 | 80.90% |
| 2 | ZC706 | VGG-8 | 4× | 1 | 84.8% | 87.6 | 14.4 | 80.90% |
| 3 | ASIC | VGG-8 | - | 2 | 6.06 mm$^2$ | 3.4 | 0.38 | 87.89% |
| | | | | CIFAR-10 HBNN Customization | | | | |
| 4 | ZC706 | VGG-8 | 1× | 1.2 | 25.4% | 18.25 | 4.3 | 85.8% |
| 5 | ZC706 | VGG-8 | 1× | 1.4 | 29.7% | 15.6 | 5.0 | 89.4% |
| 6 | ZC706 | VGG-8 | 4× | 1.2 | 100% | 73.0 | 17.0 | 85.8% |
| 7 | ASIC | VGG-8 | - | 1.2 | 2.18 mm$^2$ | 3.4 | 0.14 | 85.8% |
| 8 | ASIC | VGG-8 | - | 1.4 | 2.96 mm$^2$ | 3.4 | 0.18 | 89.4% |
| | | | | Extrapolation to MobileNet with ImageNet Data | | | | |
| 9 | ZC706 | MobileNet | 1× | 1 | 20.0% | 0.45 | 3.4 | 52.9% |
| 10 | ZC706 | MobileNet | 1× | 2 | 40.0% | 0.23 | 6.8 | 63.0% |
| 11 | ZC706 | MobileNet | 1× | 1.4 | 28.0% | 0.32 | 4.76 | 64.7% |
| 12 | ASIC | MobileNet | - | 2 | 297 mm$^2$ | 3.4 | 18.62 | 63.0% |
| 13 | ASIC | MobileNet | - | 1.4 | 145.5 mm$^2$ | 3.4 | 9.1 | 64.7% |

There have been several recent binary convolutional neural network implementations on FGPAs and ASICs that provide a baseline we can use to estimate the performance of HBNNs on ZC706 FPGA platforms (Umuroglu et al., 2017) and on ASIC hardware (Alemdar et al., 2017). The results of these implementations are summarized in rows 1-3 of Table 2. Here, unfolding refers to the number of computational units placed for each parameter, by having multiple copies of a parameter, throughput can be increased through improved parallelization. Bits refers to the level of binarization of both the input activations and weights of the network. Occupancy is the number of LUTs required to implement the network divided by the total number of LUTs available for an FPGA, or the chip dimensions for an ASIC. Rows 4-12 of Table 2 show the metrics of HBNN versions of the baseline models. Some salient points that can be drawn from the table include:

- Comparing lines 1, 4, and 5 show that on FPGA, fractional binarization offers fine-grained tuning of the performance-accuracy trade-off. Notably, a significant accuracy boost is obtainable for only slightly higher occupancy and power consumption.

- Rows 2 and 6 both show the effect of unrolling. Notably, with 1.2 average bits, there is no remaining space on the ZC706. This means that using a full 2 bits, a designer would have to use a lower unrolling factor. In many cases, it may be ideal to adjust average bitwidth to reach maximum occupancy, giving the highest possible accuracy without sacrificing throughput.

- Rows 3, 7, and 8 show that in ASIC, the size and power consumption of a chip can be drastically reduced without impacting accuracy at all.

- Rows 9-13 demonstrate the benefits of fractional binarization are not restricted to CIFAR, and extend to MobileNet in a similar way. The customization options and in many cases direct performance boosts offered by HBNNs are valuable regardless of model architecture.

## 6   Conclusion

In this paper, we present Heterogeneous Bitwidth Neural Networks (HBNNs), a new type of binary network that is not restricted to integer bitwidths. Allowing effectively fractional bitwidths in networks gives a vastly improved ability to tune the trade-offs between accuracy, compression, and speed that come with binarization. We introduce middle-out bit selection as the top performing technique for determining where to place bits in a heterogeneous bitwidth tensor. On the ImageNet dataset with AlexNet and MobileNet models, we perform extensive experiments to validate the effectiveness of HBNNs compared to the state of the art and full precision accuracy. The results of these experiments are highly compelling, with HBNNs matching or outperforming competing binarization techniques while using fewer average bits.

## Footnotes

[1]In line with prior work, we use the AlexNet model trained on the ImageNet dataset as the baseline.

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
