[Reviews · NeurIPS 2018]

Reviewer 1



The paper presents a heterogeneous binarization technique that allows every parameter in a neural network to have a different bitwidth. The authors show that the proposed method is comparable or outperforms baseline methods with fewer average bits on ImageNet with AlexNet and MobileNet models. I am not very familiar with the area, but I think this is an interesting paper that could become useful for practical applications of neural network models. While the proposed method is a small modification of previous work, the experiments are thorough and the results are quite convincing.

Reviewer 2



Deep neural networks are computationally efficient when they use only a few bits to express weights and activation. Previous work use fixed number of bits for the entire network. This paper argues that it is inefficient for some values if a small number (1 or 2) of bits is already enough. Then the paper propose to use a hybrid approach, which allows numbers to be truncated to 1, 2, or 3 bits. The method ranks all values according to their truncation errors and then decides the proportion of values using 1, 2, or 3 bits respectively. In general, values with large proportion error use more bits. The main contribution of the paper is using truncation error to decide the number of bits. In my view, the paper definitely has a contribution to the field, but the contribution is limited. The paper shows that the performance of the proposed approach with 1.4 bits is comparable to that of previous approach with 2 bit. Does the heterogeneous structure increases the implementation difficulty? Writing issues: Section 3 describes the proposed method. I propose to include then entire method in section 3. For example, equation (5) should include the entire quantization procedure and the gradient calculation.

Reviewer 3



The paper makes an observation that in general 2-bit models are more accurate than 1-bit model. The goal in this paper is to make the performance of lower-bit models closer to than of without (significantly) compromising on their efficiency. The paper achieves this by binarizing to a mixture of bitwidths as opposed to the traditional approach of a uniform bitwidth to all values. In particular, it extends the approach presented in Tang et al. (2017) that homogeneously binarizes to n bits. The paper proposes to examine individual values and decide whether to binarize them (using the Mask Tensor M). The paper proposes an algorithm to achieve an average bidwidth of B (non-integer) by taking as input a targeted distribution over individual bitwidths, P. In order to achieve this target bitwidth distribution, the algorithm greedily assigns low bitwidths to values that are closer to average values and so are likely to be accurate without employing additional bits and vice versa. The experiments are conducted using AlexNet and MobileNet using ImageNet data. The paper is well-written and easy to follow. The method is somewhat of a simple extension of the idea presented in previous work but the end-results are promising. I understand that the algorithm not only takes the desired average number of bits (B) as input but also the distribution over individual bitwidths (P). First of all, this is not clear in Algorithm 1, and should be explained. From the algorithm's description it seems that the only input is B (apart from other things). How sensitive is the algorithm to the choice of P? Ideally, the authors should have shown the variation in performance for different desired distributions (all yielding the same value of average bitwidth B). However, I am willing to listen to authors' opinion on this. Also, are there any recommendations/heuristics to choose a good P? Minor comment: the sentence in line 26 is difficult to read. I have read the author response. Thanks for your clarifications. Please include them in your next version.